# Apalutamide Prevents SARS-CoV-2 Infection in Lung Epithelial Cells and in Human Nasal Epithelial Cells

**DOI:** 10.3390/ijms24043288

**Published:** 2023-02-07

**Authors:** Amene Majidipur, Margot Morin-Dewaele, Jeanne Gaspar Lopes, Francois Berry, Julien Fouchet, Sophie Bartier, Anais Dufros Duval, Pascale Soyeux, Eric Huet, Bruno Louis, André Coste, Émilie Béquignon, Carolina Saldana, Philippe Le Corvoisier, Damien Destouches, Jean-Michel Pawlotsky, Alexandre de la Taille, Francis Vacherot, Patrice Bruscella, Virginie Firlej

**Affiliations:** 1TRePCa, Université Paris Est Créteil, F-94010 Créteil, France; 2Team “Viruses, Hepatology, Cancer“, Institut Mondor de Recherche Biomédicale, Université Paris Est Créteil, INSERM U955, F-94010 Créteil, France; 3Department of ENT and Head and Neck Surgery, Centre Hospitalier Intercommunal de Créteil, F-94010 Créteil, France; 4AP-HP, Department of ENT and Head and Neck Surgery, Centre Hospitalier Universitaire Henri Mondor, F-94010 Créteil, France; 5Team “Biomechanics and Respiratory System”, Institut Mondor de Recherche Biomédicale, Université Paris Est Créteil, INSERM U955, CNRS EMR 7000, F-94010 Créteil, France; 6AP-HP, Department of Oncology, Centre Hospitalier Universitaire Henri Mondor, F-94010 Créteil, France; 7INSERM, AP-HP, Clinical Investigation Center 1430, Henri Mondor University Hospital, F-94000 Créteil, France; 8AP-HP, Department of Virology, Centre Hospitalier Universitaire Henri Mondor, F-94010 Créteil, France; 9AP-HP, Department of Urology, Centre Hospitalier Universitaire Henri Mondor, F-94010 Créteil, France

**Keywords:** SARS-CoV-2, androgen, ARPI, TMPRSS2

## Abstract

In early 2020, the novel pathogenic severe acute respiratory syndrome coronavirus 2 (SARS-CoV-2) emerged in Wuhan, China, and rapidly propagated worldwide causing a global health emergency. SARS-CoV-2 binds to the angiotensin-converting enzyme 2 (ACE2) protein for cell entry, followed by proteolytic cleavage of the Spike (S) protein by the transmembrane serine protease 2 (TMPRSS2), allowing fusion of the viral and cellular membranes. Interestingly, TMPRSS2 is a key regulator in prostate cancer (PCa) progression which is regulated by androgen receptor (AR) signaling. Our hypothesis is that the AR signaling may regulate the expression of TMPRSS2 in human respiratory cells and thus influence the membrane fusion entry pathway of SARS-CoV-2. We show here that TMPRSS2 and AR are expressed in Calu-3 lung cells. In this cell line, TMPRSS2 expression is regulated by androgens. Finally, pre-treatment with anti-androgen drugs such as apalutamide significantly reduced SARS-CoV-2 entry and infection in Calu-3 lung cells but also in primary human nasal epithelial cells. Altogether, these data provide strong evidence to support the use of apalutamide as a treatment option for the PCa population vulnerable to severe COVID-19.

## 1. Introduction

In early 2020, the emergence of the novel pathogenic severe acute respiratory syndrome coronavirus 2 (SARS-CoV-2) in Wuhan, China and its rapid propagation (645,028,269 cases worldwide, https://www.worldometers.info/coronavirus/, accessed on 25 November 2022) posed a global health emergency. SARS-CoV-2 causes the most severe disease in older patients and in people with co-morbidities [1,2] and about 6.6 million (1.1%) cases have been fatal. In different countries hit by the pandemic coronavirus disease (COVID-19), numerous studies have reported that men were more infected than women and presented a higher mortality [3,4,5]. This male bias in mortality was even worse in areas registering a high rate of SARS-CoV-2 infection [6,7]. During previous epidemics of coronaviruses, male sex was associated with worse clinical outcomes due to severe acute respiratory syndrome coronavirus (SARS-CoV) [8] and a higher risk of dying from Middle East respiratory syndrome coronavirus (MERS-CoV) [9].

SARS-CoV-2 is a spherical or pleomorphic enveloped viral particle with a diameter of approximately 60–140 nm. The virion contains a single-stranded (positive-sense) RNA associated with a nucleoprotein within a capsid composed of structural proteins: the membrane (M), envelope (E) and spike (S) viral proteins. The spike protein is a glycoprotein composed of two subunits (Appendix A) which plays a crucial role in viral entry by binding to the cellular receptor and mediating host and viral membrane fusion [10,11]. At present, it is accepted that coronaviruses may enter the host cells via two routes: the endocytic pathway, requiring endosomal cysteine proteases activity (cathepsin B and L (Cat B/L)), and the non-endosomal/membrane fusion pathway requiring cell-surface proteases activity. For both routes, coronavirus entry into target cells depends on the binding of the S protein to a host cellular receptor. Following binding, the spike protein will be cleaved at specific sites by host proteases. This process leads to a conformational change of the spike protein, so that fusion between the virus and host membranes at the cell surface or in the endosome occurs. Hoffman, M. et al. have demonstrated that SARS-CoV-2 enters mainly via the membrane fusion pathway in lung cells and that this route relies on the presence of the angiotensin-converting enzyme 2 (ACE2) receptor and the TMPRSS2-dependent cleavage of the spike viral protein. In this in vitro study, the entry of SARS-CoV-2 into lung epithelial cells was strongly inhibited by camostat mesylate, an inhibitor of serine proteases activity [12].

TMPRSS2 is a type II transmembrane serine protease and was first identified in the context of prostate cancer (PCa) [13]. Its expression is regulated by testosterone and dihydrotestosterone (DHT) through stimulation of the androgen receptor (AR) in PCa cell lines [14]. TMPRSS2 is abundantly expressed in the luminal cells of the prostate epithelium [13]. Its expression is significantly higher in primary, advanced and metastatic PCa, suggesting a role in prostate carcinogenesis [15,16]. By using mouse models, it has been demonstrated that TMPRSS2 regulates cancer cell invasion and metastasis [16]. Lucas et al. have also demonstrated that TMPRSS2 expression is regulated by androgens in vivo and that androgen deprivation therapy (ADT) reduces, significantly, the levels of *TMPRSS2* transcripts in human PCa cells [16]. Furthermore, this regulation of *TMPRSS2* by androgens has also been observed in mice and human lung cells [17,18].

The fact that TMPRSS2 displays an important role in SARS-CoV-2 entry into host cells suggests that the modulation of its expression by anti-androgenic therapies may provide an alternative strategy to treat viral infection. Several studies explored the mechanism of TMPRSS2 regulation by AR and the possible applications for treating COVID-19. Conflicting results have been obtained regarding the effect of enzalutamide, an anti-androgen molecule approved for the treatment of castration-resistant PCa patients, on the regulation of TMPRSS2 expression. This AR inhibitor was shown to down-regulate *TMPRSS2* expression in human lung cell lines, in differentiated human airway cultures in the presence of DHT and in lung mice [19,20]. However, Baratchian et al. showed no modification of TMPRSS2 protein expression in the lungs of mice treated with enzalutamide [21]. In addition, several studies have reported conflicting effects of anti-androgens on SARS-CoV-2 entry in the lung or airway epithelial cells. Indeed, Leach et al. showed that enzalutamide, like bicalutamide, another AR antagonist used in cases of hormone-sensitive PCa patients, inhibits SARS-CoV-2 infection in ACE2-transduced A549 lung cells [19]. On the other hand, enzalutamide showed no effect on SARS-CoV-2 pseudovirus infection in human ACE2-transduced lung organoids [22] or differentiated human airway cultures [20]. Thus, the role of ADT or other androgen-targeted therapies in reducing SARS-CoV-2 infection is still debated. Indeed, if the effects seem convincing with original viruses, it is not the case with pseudoviruses. Moreover, their effects on the regulation of TMPRSS2 seem to depend on the cell type and on the presence or absence of DHT in the experiments. In order to clarify the anti-SARS-CoV-2 potency of anti-androgen molecules, we investigated the role of apalutamide, another AR inhibitor recently approved for advanced PCa treatment, during SARS-CoV-2 infection of lung adenocarcinoma Calu-3 and primary human nasal epithelial cells.

## 2. Results

### 2.1. TMPRSS2 and AR Genes Are Expressed in Lung Cells

The androgen receptor (AR) and TMPRSS2 expressions in lung cell models remain controversial. Hence, we compared TMPRSS2 and AR basal protein expressions in two lung cancer cell lines (A549 and Calu-3) versus two PCa cells lines (VCaP and LNCaP) by Western blot. Both proteins were expressed in both lung cell lines, albeit in lower amounts compared to PCa cells (Figure 1a,b).

To confirm these results obtained at the protein level, *AR* and *TMPRSS2* mRNA quantities were evaluated by RT-qPCR. While *AR* mRNA could be detected in both lung cell lines, Calu-3 cells expressed less *AR* mRNA than A549 cells (Figure 1c). *TMPRSS2* mRNA was weakly expressed in A549 cells, contrarily to Calu-3 cells. In those cells, its expression was approximately 20- to 30-fold lower than in prostate cells (LNCaP and VCaP, respectively) (Figure 1d).

These results suggest that the two lung cell lines could be used to study the AR–TMPRSS2 axis during SARS-CoV-2 infection. However, because several studies suggested that SARS-CoV-2 mainly uses the endosomal pathway in A549 cells and the TMPRSS2-dependent pathway in Calu-3 cells [12,23,24], we decided to select the Calu-3 cell line model for our study.

### 2.2. TMPRSS2 Is an Androgen-Responsive Gene in Calu-3 Cells

We next investigated the potential impact of AR signaling on TMPRSS2 expression in Calu-3 cells. Different strategies were performed to modulate the AR signaling pathway and the effect on TMPRSS2 expression was evaluated by RT-qPCR and/or by Western-blot. 

First, Calu-3 cells were stimulated for 72 h using 10 and 50 nM DHT, a 5-α-reductase metabolite of testosterone that induces efficient AR activation. DHT stimulation resulted in an augmentation of TMPRSS2 protein quantity compared to the control condition as assessed by Western blot analysis (Figure 2a). Next, Calu-3 cells were stimulated with 50 nM DHT and the kinetic of *TMPRSS2* mRNA expression was evaluated. As shown in Figure 2b, a time-dependent increase in *TMPRSS2* mRNA quantity was observed in Calu-3 cells. 

Then, the AR signaling pathway was inhibited by transfection of *AR*-targeting siRNAs in Calu-3 cells. Inhibition of *AR* expression was assessed by RT-qPCR relative to control siRNA (si ctrl). As shown in Figure 2c, we obtained 90% diminution of *AR* mRNA quantity 48 h post-treatment. Moreover, a significant decrease in *TMPRSS2* mRNA quantity was observed in *AR* siRNA-treated Calu-3 cells (Figure 2d). These results strongly suggest that *TMPRSS2* expression could be regulated by AR signaling in Calu-3 cells.

To strengthen these results, we inhibited the AR signaling pathway with a third, alternative, strategy: the use of two AR antagonists approved for PCa treatment, enzalutamide and apalutamide. Calu-3 cells were treated with different concentrations of enzalutamide or apalutamide for 72 h and the protein levels of TMPRSS2 were analyzed by Western blot. Enzalutamide treatment led to a dose-dependent decrease in TMPRSS2 protein levels (40% at 10 µM and 60% at 25 µM, respectively) (Figure 3a and Appendix A), while a non-significant modulation of TMPRSS2 protein levels was observed after apalutamide treatment at 10 µM and 25 µM (Figure 3b and Appendix A).

To verify the efficacy of these two drugs, their effect on TMPRSS2 expression was evaluated in VCaP cells. As expected, a decrease in TMPRSS2 expression was observed after enzalutamide or apalutamide treatment (Appendix A). Finally, to ensure that this decrease in TMPRSS2 expression was not due to a cytotoxic effect of enzalutamide, the viability of Calu-3 cells after enzalutamide and apalutamide treatments was tested. Importantly, the two drugs did not induce cell mortality at the tested doses (Appendix A).

All together, these results demonstrate the existence of an AR–TMPRSS2 axis signaling in Calu-3 cells. Because TMPRSS2 is an essential cellular protease for Spike processing, these results suggest that treatments with AR inhibitors could modulate the SARS-CoV-2 life cycle at the entry step.

### 2.3. AR Antagonists Impact SARS-CoV-2 Infection in Calu-3 and HNECs Cultures

Because TMPRSS2 expression is regulated by androgens in human lung Calu-3 cells and this protease is essential for SARS-CoV-2 entry, we evaluated the effect of apalutamide and enzalutamide on SARS-CoV-2 infection of human lung cells. Calu-3 cells were treated with 10 and 25 µM of apalutamide or enzalutamide for 48 h before being infected by the Wuhan ancestral SARS-CoV-2 viral strain. SARS-CoV-2 intracellular RNA quantity was evaluated 72 h post-infection by means of RT-qPCR. In the literature, enzalutamide inhibition of SARS-CoV-2 infection in lung cells has been reported [19,20,22]. Herein, in Calu-3 cells, a blockade of the androgen receptor activity by enzalutamide significantly decreased SARS-CoV-2 infection in a dose-dependent manner (Figure 4a).

Apalutamide, a recent anti-androgen presenting a good effect on progression free survival in castration-resistant PCa [25], has never been tested in the context of SARS-CoV-2 infection of lung cells. In our model, treatment with this molecule led to a drastic reduction in SARS-CoV-2 infection at 10 and 25 µM, as shown by a 90 to 95% decrease in intracellular viral RNA (Figure 4b). These results demonstrate that AR inhibition using enzalutamide or apalutamide inhibits SARS-CoV-2 infection in Calu-3 cells.

Human nasal respiratory epithelium corresponds to the initial site of SARS-CoV-2 infection. Hence, we evaluated the effect of apalutamide on SARS-CoV-2 infection using a more relevant model of uninfected primary human nasal epithelial cells (HNECs) cultivated at the air–liquid interface (ALI) [26,27,28,29]. HNECs were treated with apalutamide at 5, 10 and 25 µM for 48 h before infection with Wuhan SARS-CoV-2 and the drug was maintained for the entire duration of the experiment, i.e., for 72h until SARS-CoV-2 RNA quantification by RT-qPCR. Intracellular SARS-CoV-2 RNA was significantly decreased by approximately 75% when HNECs were treated with 10 or 25 µM of apalutamide, whereas no effect was observed at 5 µM (Figure 5).

These results demonstrate that apalutamide also inhibits SARS-CoV-2 infection in the HNECs model.

### 2.4. Apalutamide Blocks TMPRSS2 Catalytic Cleavage in Lung Cells and HNECs Cultures

Because we observed a drastic effect of apalutamide on SARS-CoV-2 infection in HNECs cultures, we evaluated TMPRSS2 expression by Western blot after treatment with apalutamide at 5, 10 or 25 µM for 48 h, mimicking TMPRSS2 protein level prior to SARS-CoV-2 infection in our previous experiment (Figure 5). In this model, two TMPRSS2 bands were observed: the zymogen (54 kDa) and the cleaved (26 kDa) forms of the protein. The zymogen form is very weakly present in HNECs, and its quantity does not seem to be affected by apalutamide treatment. However, a strong reduction in cleaved-TMPRSS2 quantity was observed during apalutamide treatment, in a dose-dependent manner (Figure 6b). A recent study raised the possibility that androgen signaling may play a role in modulating TMPRSS2 catalytic cleavage, which is necessary for its protease activity [30]. We therefore aimed to confirm the effect of Apalutamide treatment on the mechanism of TMPRSS2 proteolytic activation in HNECs and lung cells.

The cleavage fragment of TMPRSS2 was quantified by Western blot using another antibody targeting the N-terminal domain of the protein (Figure 6a) in HNECs treated with apalutamide (Figure 6c). Under these conditions, the zymogen (54 kDa) and cleaved (28 kDa) forms of TMPRSS2 were detectable. While the amount of zymogen form was not significantly altered in the presence of apalutamide, the quantity of the activated, cleaved form of TMPRSS2 decreased by approximately 20% at 5 and 10 µM and 70% at 25 µM apalutamide in HNECs (Figure 6c).

Finally, we evaluated TMPRSS2 proteolytic cleavage in Calu-3 cells by Western blot under the same treatment conditions, and by using an antibody directed against the N-terminal TMPRSS2 domain. Our results showed a decrease in the quantity of the cleaved, active form of TMPRSS2 when Calu-3 cells were treated with 25 μM apalutamide compared to DMSO (Figure 6d).

In conclusion, apalutamide treatment seems to affect TMPRSS2 activity by modulating its proteolytic cleavage in Calu-3 lung cells and HNECs cultures.

## 3. Discussion

In the present study, we have established TMPRSS2 and AR expression in Calu-3 and A-549 lung cell lines, at the mRNA and protein levels. In Calu-3 TMPRSS2-expressing cells, which are infected with SARS-CoV-2 via the “early” membrane fusion pathway, we showed that TMPRSS2 was an androgen-responsive gene. These results led us to verify the impact of anti-androgens on SARS-CoV-2 infection and more particularly of apalutamide, a molecule that had not been tested for its potential anti-SARS-CoV-2 effect up to now. Our results demonstrate that AR inhibition using enzalutamide or apalutamide leads to a significant decrease in SARS-CoV-2 infection in Calu-3 cells and that apalutamide inhibits SARS-CoV-2 infection in primary human nasal epithelial cells. While we observed weak effects of apalutamide on the inhibition of zymogen TMPRSS2 expression in HNECs or Calu-3 lung cells, this was not the case when studying the cleaved-TMPRSS2 form. We therefore hypothesize that, in addition to its effect on TMPRSS2 expression via AR, apalutamide plays a significant role on TMPRSS2 activity by modulating its catalytic cleavage in HNECs and Calu-3 cells.

At present, the number of men with severe forms or who died from the COVID-19 pandemic remains greater than the number of women [31]. One difference between sexes which can explain this severity is the immune response to infection by a more robust antiviral innate interferon response and an increased adaptive immunity towards viral antigens in women [32]. Another explanation could be the effect of sex steroids on the regulation of TMPRSS2 and ACE2 expression.

AR and TMPRSS2 proteins have been detected in lung cells, airway cells and lung tissues [17,19,21,22,30,33]. In our study, we have first evaluated the expression of these two genes at the mRNA and protein levels in two different lung cell lines: A549 cells used to analyze androgen regulation in lung cells [33] and Calu-3 cells, described as a good model for SARS-CoV-2 infection via the TMPRSS2-dependent pathway [11,34]. Although AR expression has been reported in both A549 [19,33,35] and Calu-3 cell lines [30], two separate studies were unable to detect AR in Calu-3 cells [17,22]. Our results showed lower but still detectable expressions of both AR and TMPRSS2 at the mRNA and protein levels in lung cells compared to PCa cells.

Several studies have suggested that SARS-CoV-2 mainly uses the endosomal pathway in A549 cells and the membrane fusion, TMPRSS2-dependent pathway in Calu-3 cells [12,23,24]. Thus, Calu-3 cells, which express AR and TMPRSS-2 and thereby mimic natural infection, were used for further experiments. TMPRSS2 is known to be regulated by androgens in PCa cells [15,16], but controversial observations have been reported in lung and airway cells [17,19,20,22,30,33]. Indeed, the diversity of lung cells and models that have been used adds complexity to such analyses. 

While Calu-3 cells cultured under androgen-deprived conditions with FBS-charcoal stripped serum for one week exhibited a reduction in *TMPRSS2* expression, no modification of *TMPRSS2* expression was observed with DHT stimulation [30]. In our study, we have shown an increase in *TMPRSS2* expression after DHT stimulation in Calu-3 cells. This result is not in contradiction with Treppiedi et al. [30], since they evaluated TMPRSS2 expression at 30 h post-DHT stimulation, while we observed a significant increase in TMPRSS2 expression at 72 h post-stimulation only (Figure 2b). Using siRNA directed against *AR*, the existence of an AR–TMPRSS2 axis was confirmed in Calu-3 cells. Our results are consistent with Qiao et al. and Leach et al. who have demonstrated that androgen regulates TMPRSS2 expression in lung epithelial cells [18,19].

As described for SARS-CoV-1, SARS-CoV-2 can be triggered to fuse either at the plasma membrane (“early pathway”) or at the endosome membrane (“late pathway”), depending on the availability of host proteases in its target cells [36]. Type 2 transmembrane serine proteinase (TMPRSS2) has been shown to play a major role in SARS-CoV-2 entry, by cleaving the S protein at the S2’ site (R815) [12,37,38].

Since Hoffman et al. have reported that entry of SARS-CoV-2 into host cells depends mainly on TMPRSS2 presence, several studies have investigated whether AR could be a potential target for COVID-19 therapy. In fact, retrospective surveys of PCa patients reported a potential impact of ADT on COVID-19 severity [39,40]. Specifically, Montopoli et al. found that PCa patients treated with ADT had a significantly lower risk of SARS-CoV-2 infection compared to untreated patients [39]. Thus, ADT may have a protective effect, decreasing the severity of COVID-19 in PCa patients [40]. However, other clinical studies obtained discordant results and the role of ADT in reducing COVID-19 infection severity in men is still a matter of debate [41,42,43,44,45,46]. Particularly, recent studies have shown that PCa patients under ADT treatment presented a higher risk of contracting COVID-19 and hospital admission [44,47]. Still, Schmidt et al. noted no association between ADT treatment and 30-day mortality among patients with PCa and COVID-19 [43]. Importantly, the patient population with PCa treated with ADT is statistically 70 years and older, i.e., with a high risk of developing severe COVID-19 and associated mortality. This may explain the difficulty in understanding the impact of ADT on COVID-19. 

In parallel, various studies have been carried out to evaluate the impact of AR inhibitors such as enzalutamide on SARS-CoV-2 infection, in vitro or in vivo.

Based on these experimental approaches, several studies proposed the use of androgen signaling inhibitors as a possible treatment for COVID-19. For example, Deng et al. and Leach et al. have demonstrated that enzalutamide regulates *TMPRSS2* expression and blocks viral entry in H460, A549 and H1944 [17,19]. Surprisingly, Leach et al., did not observe an effect of Enzalutamide treatment in Calu-3 cells using SARS-CoV-2 pseudoviral particles. Furthermore, another study has shown that α-5 reductase inhibitors, another class of androgen inhibitors decreasing the amounts of DHT in cells and inhibiting the AR signaling, reduced TMPRSS2 expression and viral infection in lung organoids [48]. Additionally, two studies presented contradictory results regarding the anti-SARS-CoV-2 potency of enzalutamide. In fact, no antiviral effect was observed when mouse and lung epithelial cells were infected with SARS-CoV-2 pseudovirus, nor during bronchial epithelial cell infection [20,22].

Up to now, the effect of apalutamide on SARS-CoV-2 infection has not been tested and, to our knowledge, no clinical trial has evaluated the effect of apalutamide as a potential treatment for COVID-19, as has been the case for enzalutamide [49]. Only Qiao et al. described an absence of effect of apalutamide at 10 µM on *TMPRSS2* expression, after 48 h of treatment in Calu-3 cells [18]. While we confirmed this result (data not shown), our results clearly showed a drastic effect of apalutamide pre-treatment on SARS-CoV-2 infection in Calu-3 cells, which was reproduced and confirmed in HNECs. This is a promising result because this model accurately recapitulates the initial site of infection of this virus, using the TMPRSS2-dependent (membrane fusion) pathway.

In these two models (Calu-3 and HNECs), apalutamide does not seem to drastically decrease the protein level of the TMPRSS2 zymogen form but tends to block its catalytic cleavage. TMPRSS2 is synthesized as a precursor protein (zymogen, ~54 KDa) which requires proteolytic cleavage to generate an active protease [14]. This active form is important for the fusion between SARS-CoV2 and host cell membranes via spike protein cleavage. This step is essential for the membrane fusion pathway of SARS-CoV-2 entry in host cells. Noteworthily, it was suggested that R1881, an androgen receptor agonist, could increase TMPRSS2 proteolytic cleavage [14,30]. However, the modulation of TMPRSS2 cleavages by apalutamide may not be the only mechanism explaining the effect of apalutamide on SARS-CoV-2 infection.

Indeed, Deng et al. have shown that enzalutamide treatment in mice results in the modulation of ACE2 expression [17]. In addition, when primary alveolar epithelial cells were treated with dutasteride, an α-5 reductase inhibitor, a decrease in ACE2 expression was also observed [48]. However, Treppiedi et al. did not observe an effect of androgen inhibitors on ACE2 expression in Calu-3 cells [30]. We have also investigated this hypothesis and our results did not demonstrate a significant regulation of ACE2 expression after apalutamide treatment, in both Calu-3 and HNECs models (Appendix A).

Nevertheless, our study has several limitations and raises questions that would be interesting to solve. This study would benefit from testing the anti-SARS-CoV-2 potency of apalutamide in primary human respiratory cells, such as pulmonary alveolar type II (AT2) cells which also express TMPRSS2. Additionally, the antiviral effect of apalutamide on SARS-CoV-2 variants should be investigated, as well as its potent antiviral activity against other highly pathogenic coronaviruses (SARS-CoV-1 and MERS-CoV). In this study, we have chosen to use relevant models for SARS-CoV-2 infection, i.e., not requiring ACE2 transduction and whose infection mainly occurs via the TMPRSS2-dependent pathway. However, given the drastic anti-SARS-CoV-2 effect of apalutamide, a TMPRSS-2 independent effect of this drug—still requiring the AR signaling pathway—is possible. Finally, no clinical data comparing SARS-CoV-2 infection or COVID-19 severity in patients treated with apalutamide versus enzalutamide are available.

## 4. Materials and Methods

### 4.1. Cell Culture

A549 cells (human tumorigenic lung epithelial cells, ATCC- CCL-185) were maintained in Dulbecco’s modified Eagle medium (DMEM, ThermoFischer Scientific) supplemented with 10% fetal bovine serum (FBS), 50 IU/mL penicillin, 100 µg/mL streptomycin (ThermoFischer Scientific). Calu-3 (human epithelial lung adenocarcinoma, ATCC- HTB-55) and VCaP cells (human epithelial prostate carcinoma, ATTC- CRL-2876) were cultured in A549 medium supplemented with 1% non-essential amino acids (NEAA, ThermoFischer Scientific) or 2 nM DHT (R & D Systems), respectively. LNCaP cells (human epithelial prostate carcinoma, ATCC- CRL-1740) were cultured in RPMI medium (Thermofischer) supplemented with 10% fetal bovine serum, 50 IU/mL penicillin, 100 µg/mL streptomycin and 2 nM DHT. All cells were cultivated at 37 °C, 5% CO_2_ under humidified atmosphere. Apalutamide and enzalutamide have been provided by Janssen-Cilag and Astellas Pharma, respectively. RNAi max lipofectamine (ThermoFischer Scientific) was used to transfect cells in 24-well plates following the instruction of the manufacturer and using the indicated siRNAs (10 nmol/L).

### 4.2. Preparation of Human Nasal Epithelial Cells (HNECs)

Primary HNECs were obtained from three male patients with chronic rhinosinusitis with nasal polyps undergoing ethmoidectomy, as previously described [26]. These patients gave informed consent, and the study was approved by the local Ethics Committee (Comité de Protection des Personnes IDF X 2016-01-01). The three patients benefited from an earlier 24 h negative RT-qPCR for SARS-CoV-2 RNA in nasopharyngeal swabs. 

After surgical resection in the operating room, nasal polyps were immediately placed in DMEM/F-12 supplemented with antibiotics (100 U/mL penicillin, 100 mg/mL streptomycin, 2.5 g/mL amphotericin B and 100 mg/mL gentamicin). For cellular dissociation, nasal polyps were washed with PBS (Phosphate Buffered Saline, Life Technologies, Carlsbad, CA, USA) containing 5 nM DTT (dithiothreitol, Sigma Aldrich, St. Louis, MO, USA) to eliminate mucus and blood. Then, enzymatic digestion was performed for 16 h at 4 °C (0.1% [wt/vol] pronase, Sigma Aldric, in culture medium). HNECs (10^6^ cells) were then plated in inserts (12 mm Costar Transwell, Sigma Aldrich) with 12 mm diameter polycarbonate micro-pore membranes (pore size of 0.4 µm), coated with type IV collagen (Sigma Aldrich) and incubated at 37 °C in 5% CO_2_. In the first 24 h, cells were incubated with 1 mL of DMEM/F-12-antibiotics with 2% Ultroser G in the lower chamber and DMEM/F-12-antibiotics with 10% FCS in the insert. After 24 h, the medium at the surface was aspirated and cells were washed with PBS to eliminate non-adherent cells. The culture medium (Pneumacult-ALI, StemCell Technologies, Vancouver, Canada) in the insert was removed to place the cells at the air–liquid interface (ALI). Cultures were maintained at 37 °C with 5% CO_2_. The medium in the lower chamber was then changed every day. To assess cell viability, the cells were observed daily with an optic microscope and trans-epithelial electrical resistance (TEER) was measured (Epithelial Volt/Ohm Meter 3, World Precision Instruments, Sarasota, FL, USA). As already reported, HNECs reached a stable differentiated state with the detection of ciliated, goblet and basal cells at day 21 of culture in the ALI medium [27].

### 4.3. Viruses

The SARS-CoV-2 strains used in this study were isolated from nasopharyngeal swabs from SARS-CoV-2-infected patients followed at Henri Mondor University Hospital, Créteil, France. SARS-CoV-2 viral stock was propagated in VERO-E6 cells. The viral stocks were aliquoted and stored at −80 °C. Viral titers were measured by means of RT-qPCR in VERO-E6 cells and expressed as TCID_50_ per milliliter. VERO-E6 cells were seeded in sextuplicate in 48-well plates and incubated for 2 h with 10-fold serial dilutions of viral stocks. The infectious inoculum was then replaced by the culture medium and cells were incubated for 48 h before intracellular viral RNA assessment by RT-qPCR.

### 4.4. SARS-CoV-2 Infection

SARS-CoV-2 infection of HNECs isolated from uninfected patients was performed at their apical pole with 20 µL of viral inoculum (2.04 × 10^5^ TCID_50_/mL) for 4 h at 37 °C with 5% CO_2_. HNECs were treated 48 h before infection, at the basal and apical poles, with apalutamide (Janssen) or enzalutamide (Astellas). For the Calu-3 cell line, 150,000 cells were seeded in 48-well plates 24 h prior to SARS-CoV-2 infection. Cells were infected at a final MOI of 0.1 for 2 h in DMEM 2% FBS at 37 °C with 5% CO_2_. When indicated, cells were incubated in the absence or in the presence of apalutamide or enzalutamide 48 h before infection. After infection, HNECs were rinsed at the apical pole with 200 µL PBS, while immortalized cell lines were washed and supplemented with fresh medium. When indicated, apalutamide, enzalutamide or DMSO were added at the apical pole and in the basal medium (HNECs) or the culture media (immortalized cell lines) for the duration of the experiment. Intracellular RNA was extracted at 72 h post-infection, as indicated.

### 4.5. Extraction and Quantification of Viral RNA by RT-qPCR

Intracellular viral RNA was extracted using the RNeasy Mini Kit (Qiagen). Reverse transcription was performed with the High Capacity cDNA Reverse Transcription Kit (ThermoFisher Scientific, Waltham, MA, USA). SARS-CoV-2 RNA was quantified with the TaqMan Gene Expression Master Mix (Applied Biosystems, Foster City, CA, USA) using specific primers (forward primer 5′-ACAGGTACGTTAATAGTTAATAGCGT-3′ and reverse primer 3′-ATATTGCAGCAGTACGCACACA-5′) and qPCR probe (5′-ACACTAGCCATCCTTACTGCGCTTCG [5′]Fam [3′]BHQ-1-3′). Quantitative PCR was performed with QuantStudio 5 Real-Time PCR system (ThermoFisher Scientific, Waltham, MA, USA). SARS-CoV-2 RNA levels were quantified according to the ΔΔCT method and normalized to 18S rRNA for intracellular samples.

### 4.6. Cellular RNA Extraction and Quantification by RT-qPCR

Total cellular RNA was isolated with TRIzol reagent (ThermoFischer Scientific, Waltham, MA, USA). RNA was retrotranscribed using the High capacity cDNA Reverse Transcription Kit (ThermoFischer Scientific, Waltham, MA, USA). cDNA was quantified by real-time PCR using the ROX SYBR Green PCR Master Mix (ThermoFischer Scientific, Waltham, MA, USA). Human *RPLP0* A was used as an internal control. Primers used were *RPLP0*-for 5′-ATGTGCAGCTGATCAAGACTGG-3′, *RPLP0*-rev 5′-AGGCCTTGACCTTTTCAGCAA-3′, *TMPRSS2*-for 5′-AGGAGTGTACGGGAATGTGATGGT-3′, *TMPRSS2*-rev 5′-GATTAGCCGTCTGCCCTCATTTGT-3′, *AR*-for 5′-GACTTCACCGCACCTGATG-3′, *AR*-rev 5′ CTGGCAGTCTCCAAACGCAT-3’.

### 4.7. Western Blot

Calu-3 (5 × 10^5^ cells) were seeded in 6-well plates and incubated for 24 h for adhesion. After treatments, cells were lysed in RIPA buffer. Protein samples were then denaturated at 95 °C and migrated on 10% acrylamide/bisacrylamide gel. After transfer on a PVDF membrane (ThermoFisher Scientific), blocking was performed in 5% nonfat dry milk for 1 h. Membranes were then incubated overnight at 4 °C, with primary antibodies directed against the following proteins: TMPRSS2 (ab109131, 1/1000, Abcam, Cambridge, UK), TMPRSS2 Nt (ab92323,1/1000, Abcam, Cambridge, UK), AR (sc-816, 1/1000, Santa-cruz, CA, USA), GAPDH (ab8245, Abcam, Cambridge, UK) and β-actin (ab8226, 1/1000, Abcam, Cambridge, UK). Membranes were washed in TBS-tween before incubation with the corresponding HRP-conjugated secondary antibody for 1 h. Immune complexes were detected by chemiluminescence detection with Pierce ECL Western blotting substrate (ThermoFischer Scientific, Waltham, MA, USA) using G:BOX systems (Syngene). The quantification was carried out with the software ImageJ.

### 4.8. Statistical Analysis

Statistical analyses were performed by ANOVA or t-student tests using the GraphPad Prism 9.0 software (San Diego, CA, USA). Values of *p* < 0.05 were considered significant. Results were expressed as mean ± SEM from three independent experiments.

## 5. Conclusions

In conclusion, we demonstrated that, as is the case in prostate cells, TMPRSS2 is regulated by the AR pathway in lung cells and HNECs cultures. Pretreatment with the AR inhibitor apalutamide decreased SARS-CoV-2 infection in in vitro and ex vivo models: Calu-3 cells and HNECs, respectively. Our results suggest that apalutamide’s mode of action is to modulate TMPRSS2 catalytic cleavage in Calu-3 and HNECs cultures. Enzalutamide and the more efficient AR antagonist apalutamide appear to be good candidates to limit SARS-CoV-2 infection and should be tested with local treatment in the prostate cancer population vulnerable to severe COVID-19.

## Figures and Tables

**Figure 1 ijms-24-03288-f001:**
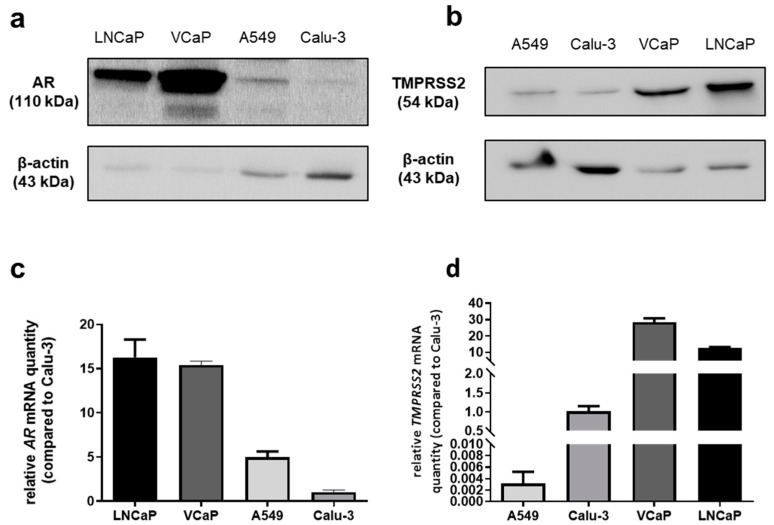
AR and TMPRSS2 protein and mRNA expressions in Calu-3 and A549 lung cells (LNCaP and VCaP PCa cell lines were used as positive controls). (**a**) Western blot analysis of AR and β-actin expression; (**b**) Western blot analysis of TMPRSS2 and β-actin expression; (**c**,**d**) mRNA expression levels of *AR* (**c**) and *TMPRSS2* (**d**) as assessed by RT-qPCR. Results were normalized to RPLP0, then to mRNA quantities in Calu-3 cells. Relative quantities are expressed as mean ± SEM of three independent experiments.

**Figure 2 ijms-24-03288-f002:**
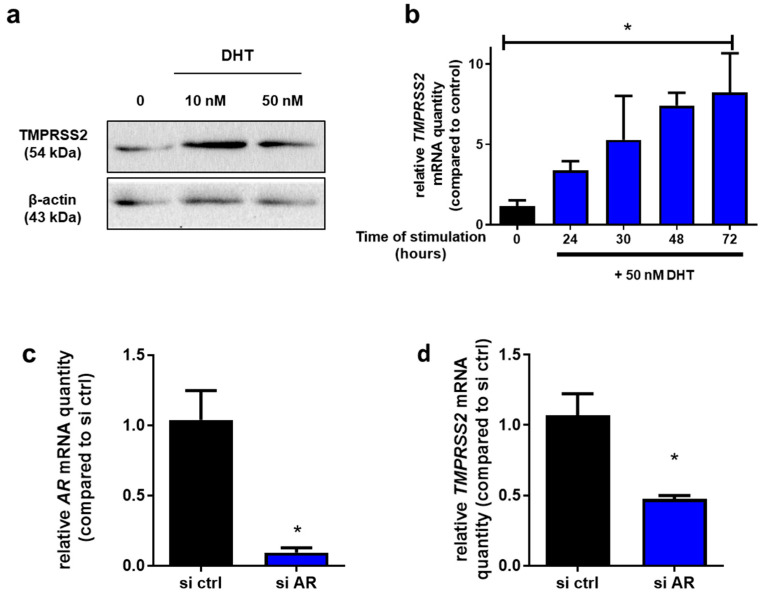
Regulation of TMPRSS2 through the AR signaling pathway. (**a**) Western blot analysis of TMPRSS2 in Calu-3 cells treated with DHT at 10 and 50 nM for 72 h; (**b**) Quantification of *TMPRSS2* mRNA in Calu-3 cells treated with DHT at 50 nM for 0, 24, 48 or 72 h by means of RT-qPCR. Results were normalized to RPLP0, then to 0 h. Relative quantities are expressed as mean ± SEM of three independent experiments (**c**,**d**) Expression of *AR* (**c**) and *TMPRSS2* (**d**) mRNA in Calu-3 cells 48 h post-transfection using siRNAs directed against *AR* in presence of DHT. Results were normalized to *RPLP0*, then to si ctrl. Relative quantities are expressed as mean ± SEM of three independent experiments. * *p* < 0.05.

**Figure 3 ijms-24-03288-f003:**
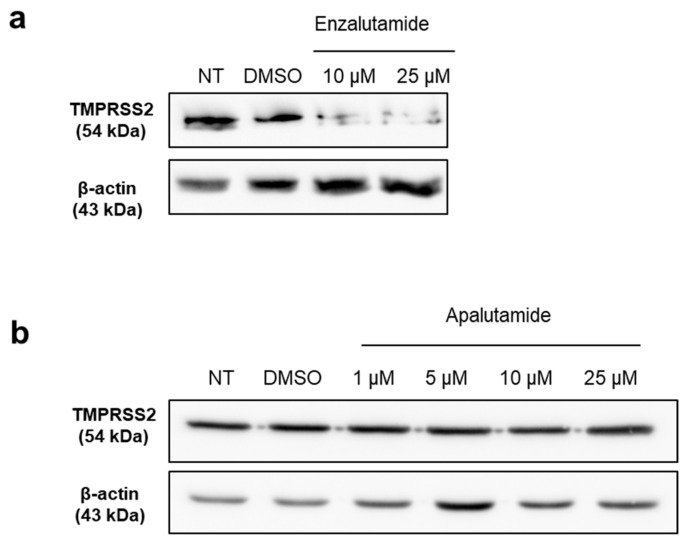
Effect of AR antagonists on TMPRSS2 expression in lung cells. (**a**) Western blot analysis of TMPRSS2 and β-actin in Calu-3 cells treated with enzalutamide at 10 or 25 µM for 72 h; (**b**) Western blot analysis of TMPRSS2 and β-actin in Calu-3 cells treated with apalutamide at 1, 5, 10 or 25 µM for 72 h.

**Figure 4 ijms-24-03288-f004:**
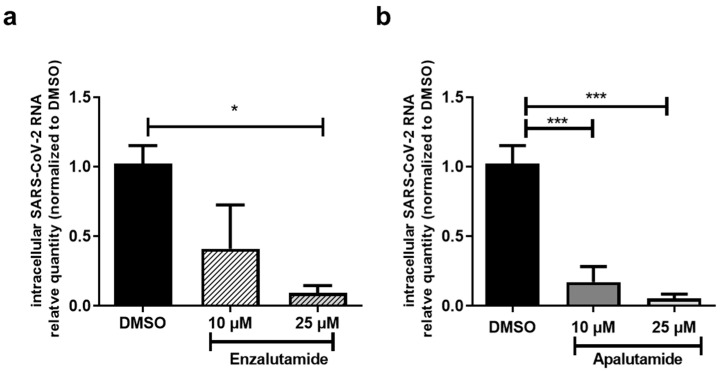
Effect of AR antagonists on Calu-3 cell infection by SARS-CoV-2. Calu-3 cells were pretreated with Enzalutamide at 10 or 25 µM (**a**) or Apalutamide at 10 or 25 µM (**b**) for 48 h before infection. Intracellular RNA was extracted 72 h post-infection. SARS-CoV-2 RNA was quantified by RT-qPCR and the results were normalized to 18S rRNA, then to control (DMSO). Results are expressed as mean ± SEM of two independent experiments. * *p* < 0.05, *** *p* < 0.001.

**Figure 5 ijms-24-03288-f005:**
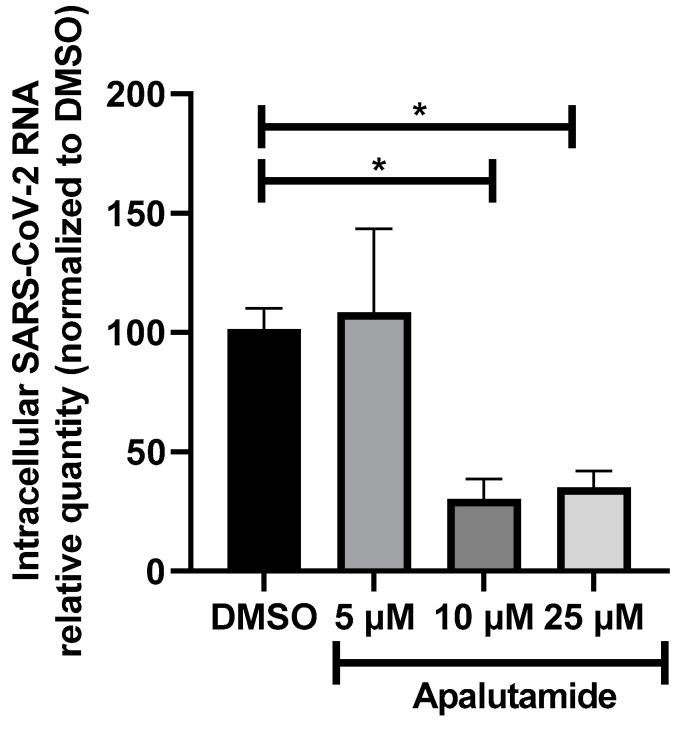
Effect of apalutamide on HNECs infection by SARS-CoV-2. Intracellular RNA was extracted from HNECs 72 h post-infection with SARS-CoV-2 (20 µL of viral inoculum, 2.04 × 10^5^ TCID 50/mL) at the apical pole. HNECS were pre-treated for 48 h before infection with DMSO (control) or apalutamide (5, 10 and 25 µM) at the basal and apical poles, and the drug was maintained for the entire duration of the experiment. SARS-CoV-2 RNA was quantified by RT-qPCR and the results were normalized to 18S rRNA, then to DMSO (control = 100%). They are expressed as mean ± SEM. (* *p* < 0.05). For each concentration of apalutamide, HNECs from at least two different patients, in duplicate, were used. Results per HNECs batch are represented in Appendix A.

**Figure 6 ijms-24-03288-f006:**
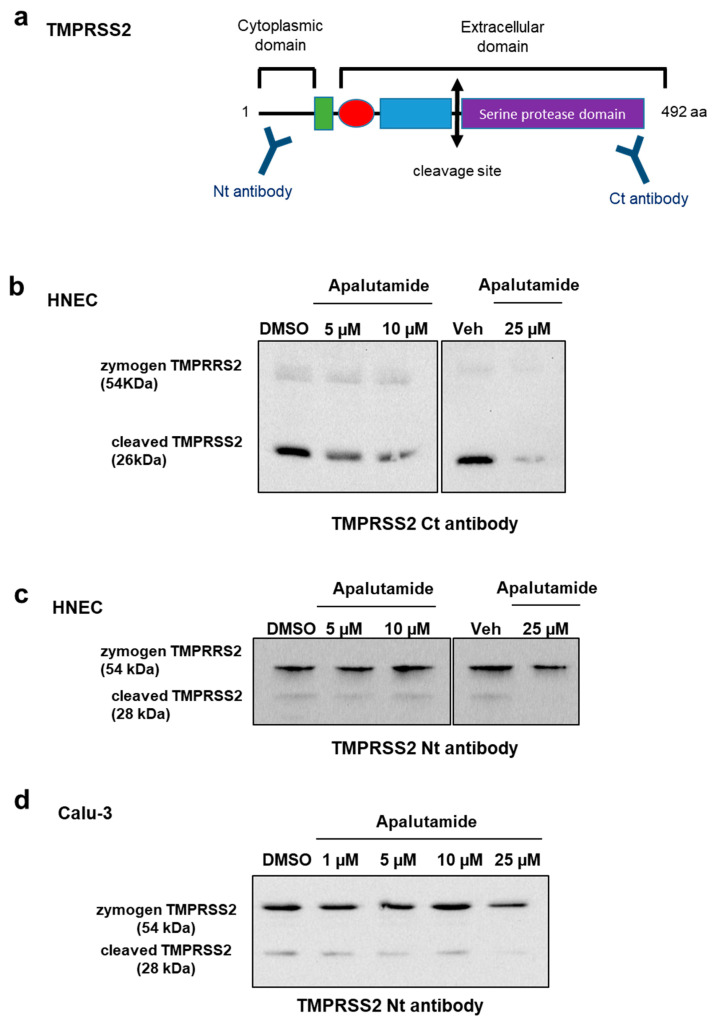
Effect of apalutamide treatment on TMPRSS2 catalytic cleavage. (**a**) Schematic representation of TMPRSS2 structure with antibodies recognition sites; (**b**–**d**) Western blot analysis of TMPRSS2 in HNECs (**a**,**c**) or Calu-3 cells (**d**) treated with apalutamide for 48 h, using an antibody directed against the C-terminal (**b**) or the N-terminal region of TMPRSS2 (**c**,**d**).

## Data Availability

The data presented in this study are available on request from the corresponding author.

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
