# Peer review of "Apalutamide Prevents SARS-CoV-2 Infection in Lung Epithelial Cells and in Human Nasal Epithelial Cells"

_ijms, 2023, doi:10.3390/ijms24043288_

Round 1

Reviewer 1 Report

The authors build a very good case for the hypothesis of AR down-regulation of TMPRSS2 and its potential role in preventing COVID19 infection.
While epidemiologic studies suggested that androgen blocking treatment is associated with lower COVID19 severity, an RCT of androgen receptor blocker has failed to show benefit among men hospitalized with severe COVID19. The presented manuscript strengthens the case for further clinical trials exploring androgen blockage, and even suggests a choice of agent and timing of AR blockade. The manuscript reads well, and the figures are clear. I have no comments.

Author Response

We are grateful to the rewiever for his/her comments on the quality of our work.

Reviewer 2 Report

The authors presented a study showing that Apalutamide prevents SARS-CoV-2 infection in lung epithelial cells and in human nasal epithelial cells. The results showed that AR antagonists appear to be good candidates for limiting SARS-CoV-2 infection and should be tested for local treatment of patients with prostate cancer vulnerable to severe COVID-19. On the whole, the work is interesting and well presented. However, the work also raises several questions and there are areas that could benefit from further clarification.

I have some comments as below.

1.     “When referring to the mRNA, the italicized notation is used”. https://www.sfedit.net/appropriate-use-of-gene-symbols-in-scientific-writing/

2.     p. 11, line 370. “All patients gave informed consent…”. Figure 5 “RNA was extracted 72 h post-infection from HNECs isolated from 2 different patients”. Please clarify the information about the number of patients from whom the cells were obtained. In the supplements, please give data for individual patients, so that the variation between patients can be seen. Were there any outliers? How were the 3 replicates from different patients averaged?

3.     P.2-3, lines 97-98 “In order to clarify the anti-SARS-CoV-2 potency of anti-androgens molecules…” Please discuss the possible reasons for the inconsistent results of the effects of anti-androgen therapy on TMPRSS2 expression and SARS-CoV-2 cell infection in the literature (some differences in the mechanisms of drug action, peculiarities of cell lines, methodological aspects, etc.).

4.     The Discussion should start with a brief summary of the main findings (Albert T.

Winning the Publications Game: The Smart Way to Write Your Paper and Get It Published. 4th ed. CRC Press; 2016). In the present study, the discussion starts with literature data.

5.     Please discuss study limitations.

Reviewer 3 Report

The study aims to explore the relationship between androgen receptor  signalling and TMPRSS2 expression in relation to infection of lung cell models with SARS CoV2. The manuscript identifies controversy in this area, with contradictory reports from previous studies, despite seemingly similar cell models and experimental design. As the information has important clinical implications and could form the basis for novel therapeutic approaches, a manuscript which addresses these inconsistencies would be valuable.  The poor quality of the some of the figures raises significant questions as to whether this manuscript is that paper. 

In reading the manuscript I was very concerned by the quality of the data and the poor labelling of figures and data. For example, but not limited to the following examples:

Figure 2 recites different timings in the legend, from those shown in the figure; please amend. 

Figure 3 is completely mislabelled, with the drugs inverted in the figure and legend. 

I am also unconvinced by the normalisation of the data in the respective fgure (Normalisation against 18S rRNA does not seem appropriate in figure 4). 

Regretfully, I can't support publication of the manuscript on this basis. 

Round 2

Reviewer 3 Report

The authors have addressed the questions I raised.